# Can Digital Economy Promote Energy Conservation and Emission Reduction in Heavily Polluting Enterprises? Empirical Evidence from China

**DOI:** 10.3390/ijerph19169812

**Published:** 2022-08-09

**Authors:** Rongwu Zhang, Wenqiang Fu, Yingxu Kuang

**Affiliations:** 1School of Management, Guangzhou University, Guangzhou 510006, China; 2College of Business, University of Houston-Victoria, Victoria, TX 77901, USA

**Keywords:** digital economy, corporate energy conservation and emission reduction, heavily polluting enterprises, green innovation

## Abstract

This paper examines the impact of digital economy on corporate energy conservation and emission reduction (CECER) using China’s A-share listed heavily polluting enterprises from 2012 to 2019 as a sample. Our results show that: (1) Digital economy can significantly increase CECER, and this effect is significant for mining and manufacturing enterprises, and less significant for power, heat production and supply enterprises; (2) Mechanism research shows that digital economy promotes CECER through enhancing the green technology innovation capability, easing the financing constraints, and boosting market competition; (3) Heterogeneity research indicates that the promotion of digital economy to CECER is more significant in economically developed regions and regions with less financial pressure from local governments. This paper clarifies the factors influencing CECER and provides empirical evidence for achieving digital economy development and government goals for CECER.

## 1. Introduction

China’s rapid economic growth in recent decades has brought serious environmental pollution [1], particularly excessive energy consumption and pollutant emissions in heavily polluting industries [2]. China’s sulfur dioxide emissions decreased from 21.18 million tons in 2012 to 4.57 million tons in 2019, and power consumption increased from 4976.2 billion kw/h in 2012 to 7486.6 billion kw/h in 2019 [3]. China has recently adopted stricter environmental supervision at the expense of economic growth, resulting in a continuous decline in pollutant emissions. However, pollution still has a negative impact on human health, and increasing energy consumption also poses a critical threat to sustainable development. Of all these pollutant emissions and energy consumptions, industrial sulfur dioxide emissions accounted for an average of 74.41% of the total, while industrial electricity consumption accounted for an average of 70.80% of the total. In view of the importance of sustainable development, China has been committed to pursuing green economy in recent years. In 2020, the Chinese government proposed the goals of reaching “peak carbon dioxide emissions” by 2030 and achieving “carbon neutrality” by 2060 [4]. It clearly points out that it is necessary to promote energy conservation and emission reduction, which is especially important in heavily polluting industries. But how to promote corporate energy conservation and emission reduction (CECER) while ensuring economic growth remains a problem demanding prompt solution. 

Existing research shows that CECER involves many factors such as government, society, capital market, and corporate executives. First, the government can guide and directly constrain CECER by formulating relevant laws, regulations and policies as a powerful external force [5,6,7] and can also supervise CECER by strengthening supervision as the main watchdog organization [8]. Second, as the public become more aware of the importance of sustainable development, they believe that enterprises should undertake more social responsibilities. Public concern about corporate pollution and moral condemnation will urge enterprises reduce pollutant emissions [9]. Third, Kong et al. [10] found that if enterprises shoulder the social responsibility of environmental protection, their market value will rise, and capital costs will reduce [11]. Therefore, investors’ preferences in the capital market will also affect corporate decisions on energy conservation and emission reduction. Finally, Zhang et al. [12] found that arrogant CEOs will increase corporate pollutant emissions, while corporate executives who have political connections will reduce pollutant emissions and increase energy conservation [1]. Although a significant amount literature has discussed the issue of corporate pollutant emissions, there are few studies from the macroeconomic perspective. How digital economy affects CECER requires further research. 

Digital economy is the product of the integration of digital technology and traditional economy [13]. It is an important opportunity for countries to achieve economic development and enhance their competitive strength in the future. It has been proven that digital economy can bring “digital dividends”, effectively reduce economic costs [13], promote regional economic growth [14] and sustainable employment [15]. The application of digital technology can effectively improve the efficiency of financial services and alleviate the problem of corporate financing constraints [16]. In addition, digital dividends can reduce corporate risks, promote corporate innovation [17], and enhance corporate value [16]. However, it is still unknown whether digital economy can promote CECER. As it is important to coordinate economic growth and environmental protection, this study is therefore of great practical significance. 

A review of the current mainstream literature indicates that existing studies have mainly focused on the development of the digital industry (Internet and ICT industry) and the digital transformation. There are three distinct views in the previous literature on the effect of the Internet and the ICT industry on environmental pollution, including: (1) the development of the Internet and ICT industry can reduce social pollutant emissions [18,19]; (2) The investment in the Internet and ICT industries has driven the development of many sub industries at the same time, causing a large amount of power and resource consumption, directly promoting the carbon emissions of carbon intensive industries such as the power sector, thus indirectly causing environmental pollution problems [20]; (3) The relationship between the two is nonlinear [21,22]. Regarding the effect of digital transformation on environmental pollution, previous literatures have reached a relatively consistent conclusion, that is, the digital transformation of the financial industry can effectively reduce pollution through technological innovation, adjustment of industrial structure and improvement of capital allocation [4,23,24]. 

Regarding the issue of energy consumption, previous studies have not yet reached a consistent conclusion. Some researchers argue that the development of the Internet and ICT industry and digital transformation increase social energy consumption by promoting economic growth [25,26]; Some other researchers argue that digital transformation improves efficiency by promoting technological innovation [27], correcting distorted industrial structures, and accelerating human capital accumulation, thereby reducing energy demand [28]. 

To sum up, most of the previous literatures investigated the impact of ICT industry or digital transformation on the environment from the macro level, and the closest research to us is Li et al. [29] and Wen et al. [24]. Li et al. [29] studied the impact of digital economy on regional environmental quality from a macro perspective (reduce PM2.5), and Wen et al. [24] studied the impact of industrial digitalization on the pollutant emission of industrial enterprises. Different from these studies, this study discusses the micro mechanism of digital economy on CECER, which is an important channel to control social pollution. This study selects the A-share listed heavily polluting enterprises in 218 Chinese cities from 2012 to 2019 as the sample and examines the effect of digital economy on CECER. Our research indicates that digital economy has a positive effect on CECER by enhancing enterprises’ green technology innovation ability, alleviating their financing constraints, and increasing market competition.

This paper contributes to the existing literature as follows: (1) Previous literature on the impact of digital economy on enterprises mainly focused on enterprise innovation [17] and enterprise value [16]. This paper focuses on enterprise environmental behavior, deeply analyzes the impact of digital economy on CECER, and provides a reference for realizing CECER. (2) This paper further discusses the specific mechanisms of the digital economy to promote CECER. The first is to enhance the green technology innovation ability of enterprises, the second is to alleviate the financing constraints of enterprises, and the third is to enhance market competition. (3) This paper considers the heterogeneous impact at the regional level and finds that digital economy can better promote CECER in areas with higher economic development level and less financial pressure from local governments. This finding helps enterprises in various regions achieve their CECER goals with more targeted information. 

The rest of this paper is organized as follows: Section 2 is the theoretical analysis and research hypothesis; Section 3 is research design; Section 4 is empirical test and analysis; Section 5 is a further study, which analyzes the mechanism and the impact of heterogeneity; Section 6 is the conclusion.

## 2. Theoretical Analysis and Research Hypotheses

Essentially, there are two ways to achieve CECER. One is improving energy efficiency by upgrading technology and optimizing technological process. The other is reducing the pollutant production and improve pollutant treatment efficiency by upgrading production and pollution treatment equipment. Theoretically, corporate technological progress and CECER involve market, capital, and corporate governance. 

First, digital economy can promote technological progress and green innovation capabilities of enterprises, thereby boosting CECER. The development of the digital economy is accompanied by the technological change of industry 4.0, which represents the current development trend of manufacturing automation technology. Technology such as artificial intelligence enable traditional manufacturing to transform into intelligent manufacturing [30], greatly improving the production efficiency of the manufacturing industry. With these emerging technologies, enterprises can upgrade production and pollution treatment equipment and optimize technological processes, thereby improving energy efficiency and reducing pollutant emissions. A large number of engineering studies have shown that smart factories, a product of digital economy, prompt enterprises to enhance electricity efficiency through the application of technologies such as machine learning and big data [31], thereby reducing the energy consumption of enterprises and the generation of pollutants. In addition, open-source technology brought by the digital economy can reduce R&D costs of enterprises, thereby enhancing the green technology innovation capability of enterprises and promoting CECER. 

Second, digital economy can ease the financing constraints of enterprises. As a result, enterprises can invest more in environmental protection. Enterprises need capital to replace or upgrade production and pollution treatment equipment to improve energy efficiency and reduce pollutant emissions. And many evidences show that financing constraints is a major obstacle to CECER [32,33]. Digital economy has given birth to new financial models such as Internet finance through technology such as big data. Its role lies in: (1) reducing the external financing cost of enterprises by promoting the development of the traditional financial industry and alleviating the financing constraints of enterprises [34]; (2) Through the progress of information and communication technology, the problem of information asymmetry between investors and enterprises can be greatly reduced. With advanced information and communication technology, investors and enterprises can communicate directly, thereby alleviating information asymmetry and reducing transaction costs; enterprises can implement direct financing, thereby attracting more investment [23] and effectively easing corporate financing constraints. New financial models spawned by digital economy prompt banks to develop new credit technologies that can expand the coverage of bank financial services [35], thereby increasing the supply of credit to enterprises. Financial technology brought by digital economy also promotes the development of shadow banking [36], which further facilitates the development of the corporate bond market [37], thereby easing corporate financing constraints. 

Third, the development of the digital economy can boost market competition. Due to fierce market competition, enterprises will invest more in environmental protection and strengthen corporate governance, so as to avoid excessive pollution caused by agency problems [38] and personal decisions made by corporate management [12]. Digital economy has spawned new technologies and new models, creating new demands and changing supply markets. For example, virtual reality technology has spawned many VR experience stores; Artificial intelligence and Internet of Things technologies boost demand for smart home appliances and smart cars, and urge home appliance and automobile manufacturers to enhance market competitiveness through digital transformation. In other words, digital economy will further change the product market structure and boost market competition, and changes in market competition will in turn affect corporate decisions [39,40]. Facing stricter environment supervision and fiercer market competition, heavily polluting enterprises will proactively invest more in energy conservation and emission reduction to meet government goals, get more government subsidies [1] or attract investors [10,11], thereby ensuring sustainable growth. From another perspective, intensified market competition drives up the prices of factors of production [41]. Cost pressure not only forces enterprises to improve energy efficiency, but also encourages enterprises to reduce excess production and thus reduce pollutant emissions. Besides, in a highly competitive environment, competitors and the media will hype excessive pollutant emissions, so heavily polluting enterprises will carefully consider energy consumption and pollutant emissions. 

It is worth noting that market competition is a powerful corporate governance mechanism in China [42]. State-owned enterprises belong to all citizens in terms of property rights, and the state, which is entrusted by all citizens, designates government officials as chairmen to manage state-owned enterprises. Owner absence is easy to occur in state-owned enterprises due to multi-layer principal-agent, and neither salary incentives nor debt constraints can effectively stimulate managers to work hard. Therefore, the bankruptcy risk brought by market competition can motivate the managers of state-owned enterprises, thereby improving corporate governance. As there is no cushion in place for non-state-owned enterprises, market competition brings greater operational risks for non-state-owned enterprises, so market competition can be an effective supplement to corporate governance [43]. And an effective corporate governance can avoid excessive corporate pollution caused by personal decisions made by corporate management such as CEOs [12]. At the same time, it can also avoid the agency problem hindering the enterprise’s emission reduction and environment investment [38]. Therefore, market competition, as an external factor, can effectively promote CECER. 

Fourth, digital economy can alleviate information asymmetry, make corporate pollution more transparent, and realize CECER. Zhang et al. [44] found that information asymmetry has become an impediment to environmental governance as it increases the cost of environmental governance and weakens the effectiveness of environmental policies. Digital economy enables the rapid and widespread dissemination of corporate information by promoting the advancement of information and communication technology, thereby effectively alleviating information asymmetry [45]. And digital governance based on 5G and big data, regardless of time and space limits, strengthens the information exchange between the government, the public and enterprises through the Internet, which effectively alleviates the lack of regulators, placing corporate pollutant emissions under more effective supervision. In addition, El Ghoul et al. [11] found that if enterprises shoulder the social responsibility of environmental protection, their value will rise, and equity financing costs will reduce [10], which shows that investors prefer clean production. To maximize value, enterprises will devote more energy to promote CECER and inform investors of their decision through the Internet. From another perspective, the public is becoming more and more concerned and sensitive about corporate pollution. Krüger [46] also found that investors respond more to negative events related to corporate social responsibility than positive events, so alleviated corporate information asymmetry makes corporate pollutant emissions more transparent, and public concern about environment urges enterprises to reduce pollutant emissions [9]. Therefore, alleviating information asymmetry can effectively play the role of external supervision on CECER. 

Therefore, based on the above analysis, we propose the following research hypotheses.

**H1:** 
*Digital economy can significantly promote CECER.*


**H2:** 
*Digital economy promotes CECER by enhancing the green innovation capabilities of enterprises, alleviating corporate financing constraints, and boosting market competition.*


## 3. Methodology and Data Description

### 3.1. Sample Selection and Data Sources

Our sample consists of enterprises listed in Shanghai and Shenzhen A-share stock markets from heavily polluting industries in 218 cities in China. According to the “List of Listed Companies’ Environmental Protection Inspection Industry Classification Management” issued in 2008 by the Ministry of Ecology and Environment of People’s Republic of China, heavily polluting industries include thermal power, steel, cement, electrolytic aluminum, coal, metallurgy, chemical industry, petrochemical, building materials, papermaking, brewing, pharmaceuticals, fermentation, textile, tanning and mining, a total of 16 industries. The sampling period is 2012–2019 due to the data availability. Energy consumption and pollutant emissions of heavily polluting industries account for a large proportion of the total [2], and are highly representative. Following Kong et al. [47], initial data are processed as follows: (1) Exclude enterprises with negative total assets, negative current liabilities and leverage ratios exceeding 100%; (2) Exclude enterprises where important variable data are missing; (3) Exclude ST firms. Our final sample includes a total of 6203 enterprise-year observations. Data of the explanatory variable digital economy and provincial and municipal GDP data come from the China City Statistical Yearbook and the Digital Financial Inclusion Index of China compiled by Guo et al. [48], respectively, and other variable data come from the China Stock Market & Accounting Research Database. To eliminate the influence of extreme values, all variables are winsorized on the 1% and 99% quantiles.

### 3.2. Model Specification and Variable Definition

Considering the lagged effect of macroeconomic development on enterprises and the endogeneity of the model, explanatory variable and control variables lagging one phase are used in this paper. To estimate the relationship between digital economy and CECER, the following fixed effects model is used for empirical analysis: (1)$$ER_{i,t}=\alpha_{0}+\alpha_{1}Dige_{i,t-1}+\overset{n}{\underset{k=2}{\sum }}\alpha_{k}Control_{i,t-1}+Fixed+\varepsilon$$

Among them, $ER_{i,t}$ is the energy conservation and emission reduction of enterprise *i* in period *t*. Lack of energy consumption and pollutant emission data of micro-enterprises has brought certain obstacles to related research, while the method of text mining can well reflect whether an enterprise has implemented energy conservation and emission reduction. Therefore, following the research design of Kong et al. [1], this study uses the frequency (logarithmic) of energy conservation and emission reduction-related terms as a proxy variable for corporate energy conservation and emission reduction by analyzing the annual reports of listed enterprises. The terms are determined based on the “Decision of the State Council on Strengthening Energy Conservation”, “The 13th Five-Year Plan for Energy Conservation and Emission Reduction” and the “14th Five-Year Plan for Energy Conservation and Emission Reduction” issued by the Chinese government. Specific terms are “energy conservation”, “emission reduction”, “pollution reduction”, “consumption reduction”, “low carbon”, “carbon reduction”, “saving”, “recycling”, “low emission”, “low energy consumption”, “sustainable”, “renewable”, “clean production”, “clean and efficient”, and “resourceful utilization”.

The explanatory variable $Dige_{i,t-1}$ represents the digital economy development level of the *i* region in the *t* − 1 period. This paper follows the method of Zhao et al. [49], and measures from the two aspects of Internet development and digital inclusive finance and the five indicators of Internet Penetration Rate, Number of Related Employees, Related Output, Mobile Phone Penetration Rate, and Digital Financial Inclusion Index [48]. Internet Penetration Rate measures the number of internet users. Number of Related Employees includes employees in information transmission, computer services and software. Related Output measures the total revenue of the telecommunication business. Mobile Phone Penetration Rate measures the number of cell phone subscribers. Digital Financial Inclusion Index compiled by Guo et al. [48] is constructed from three dimensions: the breadth of digital finance coverage, the depth of digital finance use, and the degree of digitalization, with a total of 33 specific indicators. This paper standardizes the data of the five indicators with principal component analysis, reduces the dimension, and finally gets a comprehensive index of the digital economy development level. $\alpha_{1}$ in Model (1) is the main observation coefficient in this paper.

Control represents a string of control variables. This paper mainly controls the factors of enterprise level and regional economic development. The variables at the enterprise level are corporate size (Size), firms’ state ownership (SOE), sale growth (Sale), return on assets (ROA), operating cash flow (OCF), book-to-market ratio (BM), CEO duality (Duality), ownership concentration (Top5), and equity balance (Top5_1). This paper controls both enterprise-individual and year fixed effects and the standard errors of all regression results at the city level are clustered, and the specific variable definitions are shown in Table 1.

## 4. Empirical Test and Analysis

This section may be divided by subheadings. It should provide a concise and precise description of the experimental results, their interpretation, as well as the experimental conclusions that can be drawn.

### 4.1. Summary Statistic

Table 2 presents descriptive statistic of all variables. It can be found that the mean value of ER is 3.318, the minimum and maximum values are 0.693 and 5.416, indicating that there is a large difference in CECER during the sampling period. The minimum value of digital economy (Dige) is −0.95, the maximum value is 8.514, and the standard deviation value is 1.954. Both the difference between the maximum value and the minimum value and the standard deviation are large, indicating that different regions differ greatly in the level of digital economy. The median value of Dige is 0.276, which is smaller than the mean value of 0.946, indicating that the level of digital economy in the regions where more than half of the sample enterprises are located is lower than average, and region digital economy development is unbalanced in China. 

### 4.2. Baseline Results

Table 3 reports the estimated results of Model (1). Column (1) reports the results without any control variables. Column (2) reports the results with control variables. All coefficients of Dige are positive at the 1% significance level, the research hypothesis H1 of this paper is verified. In addition, according to the “Guidelines for Industry Classification of Listed Companies” revised and issued by the China Securities Regulatory Commission in 2012, the heavily polluting enterprises in this paper belong to the mining industry (B), the manufacturing industry (C), and the electricity and heat production and supply industry (D). Accordingly, this paper further examines the effect of digital economy on CECER in different polluting industries. Columns (3)–(5) are the results of categorical regression on industry categories. It can be found that for enterprises in the mining industry (B) and manufacturing industry (C), the estimated coefficients are all positive at the 1% significance level, indicating that digital economy can significantly promote energy conservation and emission reduction in mining and manufacturing enterprises; However, for enterprises in the electricity, heat, gas and water production and supply industry (D), the estimated coefficients are not significant, indicating that digital economy cannot significantly promote energy conservation and emission reduction in these enterprises. The possible reason is that these enterprises are generally energy providers assigned by the government, with the aim of ensuring energy supply and maintaining social stability. As China’s economy grows rapidly, the demand for resources increases, forcing these enterprises to increase their capacity and pollutant emissions. 

### 4.3. Tackle Endogeneity

#### 4.3.1. Instrumental Variable

To investigate the effect of digital economy on CECER, this study uses the main explanatory variable and all control variables lagging one phase in Model (1) to alleviate endogeneity. However, the empirical results may still be affected by some unobservable factors or the omission of important variables, resulting in the deviation of the estimation coefficient. Therefore, we further adopt instrumental variables to solve these problems. Following Chong et al. [50], this paper uses the mean digital economy development level of other prefecture-level cities in the same province and the communication data of each city in 1984 as instrumental variables, and adopts IV-GMM method for test. 

The digital economy of prefecture-level cities in the same province is interconnected to a certain degree due to the regulation of market mechanism or higher-level government. And due to the limitation of jurisdiction, prefecture-level cities have no right to deal with pollution discharge of polluting enterprises in other cities, so exogeneity conditions are met. In addition, digital economy is based on modern Internet technology, and historical communication infrastructure represents the level and demand of local communication technology, exerting an important impact on future application and development of communication technologies such as the Internet. Therefore, correlation conditions are met. However, historical communication data can hardly impact energy consumption and pollutant emissions of enterprises decades later, so exogeneity conditions are met. It should be noted that, since the communication data of each prefectual-level city in 1984 is in the form of cross-section, this paper follows the method of Nunn and Qian [51] to construct the interaction term between the number of Internet users in each year and the number of post offices in each city in 1984 and generate time-varying panel data. 

Table 4 reports the test results of the instrumental variables and the estimated coefficients of the regression variables. The test results of instrumental variables show that the instrumental variables selected in this paper are very effective, and the regression results of the second stage show that the effect of digital economy on CECER is still significantly positive, and the estimated coefficient improves, indicating that the conclusions of this paper are robust. 

#### 4.3.2. Differences-in-Differences (DID)

This paper further mitigates potential endogeneity by differences-in-differences. In 2016, several regions in China were approved to establish national big data comprehensive experimental zones (Guizhou, Beijing Tianjin Hebei, Guangdong, Shanghai, Henan, Chongqing, Shenyang and Inner Mongolia respectively), aiming to use digital technology to create the whole governance chain of “digital government”, the whole industry chain of digital economy and the whole service chain of “digital livelihood”, and promote the deep integration of “digital government”, digital economy and “digital livelihood”, as well as economic transformation and upgrading. To implement this policy, local governments have tried their best to promote regional digital economy development from the perspectives of digital industry talents, digital industry infrastructure, and key digital innovation-based enterprises. Based on this, this paper constructs the following differences-in-differences model to examine the effect of regional digital economy development on CECER: (2)$$ER=\beta_{0}+\beta_{1}Treat\times Post+\overset{n}{\underset{l=2}{\sum }}\beta_{l}Control+Fixed+\varepsilon$$

Among them, *Treat* is 1 if the city or province belongs to the big data comprehensive experimental area, otherwise it is 0. *Post* is 1 if the year is 2016 and later, and 0 before 2016. The meanings of other variables and fixed effects remain unchanged. The coefficient $\beta_{1}$ of *Treat* and *Post* is the key observation in this paper, which measures the impact of policies promoting the development of the digital economy on CECER. In addition, this paper takes the control variable as the matching variable, and uses the neighborhood matching method with a caliper of 0.05 to perform 1:1 propensity score matching (PSM) and select the control group for regression test [52]. With these efforts, the problem of sample selection is solved.

In addition, this paper uses the method of Hering and Poncet [53] to conduct a parallel trend test by examining the annual effect. Specifically, dummy variables is constructed with 2016 (Current) as the cut-off point, 2013, 2014 and 2015 are the third year (Pre3), the second year (Pre2) and the first year (Pre1) before policy implementation, respectively, while 2017, 2018 and 2019 are the first year (After1), the second year (After2) and the third year (After3) after policy implementation, respectively. Then the experimental group variable (Treat) is multiplied with the dummy time variables, the test results are shown in Table 5. 

Columns (1) and (2) of Table 5 are the differences-in-differences estimation results. The estimated coefficients of Treat×Post are all positive at the 1% significance level, indicating that the regional digital economy development policy has significantly promoted CECER, which is consistent with the expected results of this paper. Besides, the parallel trend test in Column (3) shows that there is no significant difference between the experimental group and the control group before policy implementation, and the estimated coefficients for three years after policy implementation are continuously significantly positive, which further proves that the digital economy can promote CECER in the long term. 

### 4.4. Other Robustness Tests

To ensure the reliability of the results, this paper conducts the following robustness tests. 

Exclude the influence of policy factors. Since corporate pollutant emissions are largely constrained by national policies, this paper further excludes the impact of relevant policies on the empirical results. In 2016, the State Council of China promulgated the “Implementation Plan for the Permit System for Controlling Pollutant Discharge”. This policy details the requirements for the discharge of various corporate pollutants and emphasizes the need to limit the total discharge of corporate pollutants to improve the environment. It also asks governments at all levels to tighten control on corporate pollution discharge and pollution discharge permits. Since the policy was introduced in November 2016, this paper believes the policy didn’t produce an effect until after 2017. The regression test is conducted after the interaction term between policy dummy variable (policy is 1 when the year is greater than or equal to 2017, otherwise 0) and explanatory variable is added to Model (1). This paper also excludes samples after 2017. The test results in Columns (1) and (2) of Table 6 show that the estimated coefficient of the core explanatory variable Dige is still significantly positive, and the results are still robust after policy factors are excluded. 

Take the impact of regional and industry factors into account. Both digital economy and CECER are affected by various factors at the regional level. At the same time, industrial features also affect the CECER. Therefore, this paper further controls the city and industry fixed effects. The results in Column (3) of Table 6 show that the conclusions of this paper are robust after regional and industry factors are taken into account. 

Change the clustering method. To test the impact of different clustering methods on the results, this paper further adopts corporate level clustering and double clustering by city and year. Columns (4) and (5) of Table 6 are the test results of using corporate level clustering and double clustering of city and year respectively, which indicate that the estimated coefficient of Dige is still significant. 

Substitute the explained variable. Sulfur dioxide emissions can truly reflect corporate energy efficiency and emission reduction [54]. Therefore, this paper collects and matches the sulfur dioxide emissions of sample companies from 2012 to 2019. As the pollutant emission data of micro enterprises in China are disordered, incomplete or incompletely disclosed, only 323 sample observations are obtained in this paper, which are used as explained variables and included in Model (1) for re-testing. The test results in Column (6) of Table 6 show that the estimated coefficient of Dige is significantly negative, indicating that digital economy has effectively reduced sulfur dioxide emissions from polluting enterprises and improved energy efficiency. In addition, there may be some noise in the method of text mining. To avoid the interference of the industry idiosyncratic nature of the selected keywords, this paper subtracts the average ER value of the industry in the current year from the ER value of each enterprise, thereby eliminating the common measurement deviation at the “industry-year” level. The results in Column (7) of Table 6 show that the results remain unchanged after re-measurement of the explained variables.

In addition, we consider the impact of two relevant market forces: Environmental, Social, and Governance (ESG) investment and quantitative easing (QE) policies [55]. First, the sample period of this paper coincides with the massive increase in popularity of ESG investing [56,57], The increase in the fraction of institutional investors that follow ESG policies is likely correlated with the digitalization trends in China. By construction, as more institutional investors incorporate ESG aspects into their portfolio allocation decisions firms become more environmentally friendly due to investor’s preferences and improve CECER. Second, in response to the financial and sovereign-debt crises that occurred in the late 2000s and early 2010s, central banks of major economies (e.g., US Federal Reserve, the European Central Bank, the Bank of England, and the Bank of Japan) engaged in massive quantitative easing policies [55,58]. Affected by the 2008 financial crisis, the Chinese government implemented the “4 trillion economic stimulus plan” in November 2008. However, this QE policy was not in the sampling period of this study. China did not propose any new QE policy after 2008, Therefore, we use US Federal Reserve (FED) quantitative easing policies to measure its effect on CECER. FED QE policies have increased the liquidity of capital, and this injection of liquidity may have contributed to increased cross-border capital flows, reducing the cost of capital of the Chinese companies of the study, and making their investor clientele comprised of a higher fraction of global institutional investors who value ESG investments and CECER.

Following Hattori et al. [58] and Cortes et al. [55], we control the impact of QE policies using event-study approach. Specifically, we set dummy variables based on the timing of the FED’s QE policy. In addition, we use enterprise ESG ratings as a proxy for ESG investment due to the unavailability of corporate ESG investment data. The rationality of this approach is that only companies that invest more in ESG can obtain better rating. ESG rating scores are measured by the ESG index developed by Sino-Securities Index Information Service (Shanghai, China) Co. Ltd., which is widely recognized by the industry and academia [59]. As shown in Table 7, our core explanatory variables are still significantly positive after incorporating QE policy and ESG rating into the benchmark model for testing.

Finally, this paper considers the effect of omitted variables. We follow the method of Oster [60] to estimate the magnitude of potential estimation biases. Under the condition of δ=−1, we estimate the value range of β by changing the value range of *R_max_* ($R_{max}\in \left(R,1\right]$) If the value range of β does not include 0, it means that the coefficient is stable.

As shown in Table 8, the value range of Beta is 0.02407 to 0.05098, excluding 0, indicating that our results are robust, and the core conclusions of this paper are not too biased due to the problem of missing variables.

## 5. Further Research

### 5.1. Analysis of the Impact Mechanism

This section will further discuss the mechanism through which digital economy impacts CECER. According to the previous analysis, digital economy can promote social and technological progress, which is largely reflected at the enterprise level, and technological progress facilitates CECER. Digital economy can also reduce economic costs and promote the development of the financial industry, thereby reducing the cost of external financing for enterprises. And the advancement of digital technology can expand the coverage of banking financial services, lower the threshold of the financial industry, attract more investors, and further ease the financing constraints of enterprises, providing more financial support for enterprises to invest in environmental protection. Finally, the development of the digital economy has created many new demands and “blue ocean” fields, which will further change the product market structure and boost market competition. Fierce market competition forces enterprises to devote more energy to promote CECER, and effectively improve corporate governance, avoiding excessive pollution caused by internal corporate governance. 

This paper uses the following model to test the mechanism through which digital economy impacts CECER.
(3)$$M_{i,t}=\gamma_{0}+\gamma_{1}Dige_{i,t-1}+\sum_{m=2}^{n}\gamma_{m}Control_{i,t-1}+Fixed+\varepsilon$$

Among them, *M* represents three variables that measure the impact mechanism, namely, corporate green innovation (Innovation), financing constraints (SA), and market competition (Compete). This paper uses the number of green patent applications (take the logarithm) to measure the green technology innovation capacity of enterprises, and the absolute value of SA index to measure the financial constraints faced by enterprises. The absolute value of SA is positively correlated with corporate financing constraints. Finally, following the methods of Zhang et al. [39] and Giroud and Mueller [61], this paper uses the HHI index to measure the degree of market competition, and excludes the ST enterprises and the newly listed enterprises, delisted enterprises, or enterprises whose delisting is suspended. The main focus of this paper is the coefficient $\gamma_{1}$. The empirical results are reported in Table 9. 

The estimated coefficient of Dige in Column (1) is significantly positive at the level of 5%, indicating that digital economy boosts the green innovation capabilities of enterprises, facilitating CECER. The coefficient of Dige in Column (2) is significantly negative, indicating that digital economy significantly reduces the financing constraints of polluting enterprises. Polluting enterprises lifted out of financing constraints can invest more in environmental protection, and replace or upgrade production and pollution treatment equipment, thereby effectively saving energy and reducing pollutant emissions. The estimated coefficient of Dige in Column (3) is significantly negative, indicating that digital economy significantly boosts market competition and improves corporate governance, thereby avoiding excessive pollution caused by agency problems and personal decisions made by corporate management. Besides, facing fierce competition, polluting enterprises that seek sustainable growth will try to strengthen ties with the government through cleaner production and meet the preferences of investors. Therefore, the research Hypothesis H2 is confirmed. 

### 5.2. Heterogeneity Analysis

Government subsidies for environmental protection investment is one of the important driving forces for enterprises to save energy and reduce emissions [62]. Enterprises which located in underdeveloped regions receive less subsidies and are less motivated to save energy and protection the environment. At the same time, government regulation helps control the excessive energy consumption and pollutant emissions of enterprises [8]. With weak environmental regulation, less developed regions may not be able to effectively manage these problems [63]. In addition, the digital economy will also generate unbalanced development based on regional economic endowments. Backward digital economy in regions with poor economic endowments may not have a significant effect on CECER. 

Facing great financial pressure, local governments will provide less environmental protection subsidies, and ease environmental regulation due to political assessment. In other words, local governments will pursue economic growth at the expense of the environment, devoting less energy to environmental regulation. At the same time, the financial pressure on local governments will be borne by local enterprises to a certain extent. For example, local governments will increase the actual tax rate of enterprises [64], increasing the burden on enterprises. This will ultimately affect the effect of digital economy on CECER. 

To examine the impact of local economic development level and government financial pressure, this paper divides the sample into two groups: enterprises in developed regions and enterprises in less developed regions, according to the GDP of each city. The ratio of fiscal gap to fiscal revenue of local governments is used to measure the financial pressure faced by each city. Fiscal gap is the general public budget expenditure minus the general public budget revenue. The larger the value, the greater the financial pressure on local governments. The sample is divided into two groups according to the median financial pressure: higher financial pressure group and lower financial pressure group, and regression test is conducted after grouping. 

The estimates reported in Table 10 show that the Dige coefficient in Column (1) is significantly positive, while the Dige coefficient in Column (2) is not significant, which indicates that digital economy can promote energy conservation and emission reduction of polluting enterprises in developed regions. In addition, the results in Column (3) are not significant, while the results in Column (4) are significantly positive at the 1% level. This shows that the digital economy can significantly promote CECER only when the financial pressure of local governments is small.

## 6. Conclusions

Digital economy is currently an important force driving economic growth, while corporate energy conservation and emission reduction helps achieve green economic growth. Whether the two can be effectively integrated is still a major issue that needs to be answered urgently. This paper uses the A-share listed heavily polluting enterprises in 218 Chinese cities from 2012 to 2019 as the sample and examines the effect of digital economy on CECER. Major findings of this paper are: First, digital economy has a significantly positive effect on CECER. It is found that this effect is more significant for mining and manufacturing enterprises. After controlling endogeneity and conducting a series of robustness tests, the conclusion of this paper still holds. Second, the mechanism study shows that digital economy promotes CECER through three mechanisms: one is enhancing the green technology innovation capability of enterprises, which provides an important technical foundation for enterprises to implement energy conservation and emission reduction; the other is easing the financing constraints of enterprises, so that enterprises can invest more in environmental protection; The third is boosting market competition. Due to fierce market competition, enterprises will invest more in energy conservation and emission reduction to meet government goals and investor preferences, ensuring sustainable growth. In addition, market competition, as one of the important means of corporate external governance, avoids excessive pollution caused by agency problems and personal decisions made by corporate management. Third, the heterogeneity test shows that the digital economy can significantly promote energy conservation and emission reduction of enterprises in economically developed regions and regions with less financial pressure from local governments. 

This study enriches the theoretical research on the corporate behavior of energy conservation and emission reduction by further revealing factors that affect CECER. It also provides reliable empirical evidence and useful policy implications. First, the development of digital economy is not only an important force for national economic growth, but also can promote CECER to achieve the goal of green and sustainable economic growth. The impact of COVID-19 pandemic has further highlighted the importance of the digital economy. During the critical period of the development of the digital economy, governments should increase policy support, encourage the digital transformation of traditional industries such as manufacturing, help digital technology companies to accelerate business growth, and actively cultivate digital technology talent. Second, based on regional economic endowments, the development of the digital economy is highly uneven geographically, which weakens the energy-saving and emission-reduction effects of enterprises. It is necessary to introduce digital economy development assistance policies for backward regions, promote the experience of digital economy construction in economically developed regions to backward regions, strengthen exchanges in digital economy construction between regions, and promote the balanced development of the national digital economy. These policies will help alleviate the status quo in backward regions where environmental protection are sacrificed for economic growth.

Our study has the following limitations. First, the digital economy is a newly developed concept, The measurement of the digital economy could be further improved. Second, our research sample comes from China, and China is in a critical period of coordinating high-speed economy, improving people’s livelihood, and environmental protection, Therefore, our findings have reference significance for other developing countries but may not be able to generalize to developed countries.

Scholars should further develop more accurate digital economic indicators. In addition, the development of the regional digital economy is mainly led by the government, causing a gap between the development level of the digital economy in the region and corporate performance in the digital economy. This asymmetry may mean the waste of public resources. Future research may be performed to examine economic consequences of this asymmetry and ways to overcome the problem.

## Figures and Tables

**Table 1 ijerph-19-09812-t001:** Definitions of related variables.

Variable Name	Variable Symbol	Definitions of Variables
Corporate energy conservation and emission reduction	ER	The natural logarithm of the frequency of energy conservation and emission reduction related words in corporate annual report.
Digital economy	Dige	The development level of urban digital economy.
Corporate size	Size	The natural logarithm of corporate total assets in China Yuan (CNY) at the end of the period.
Sale growth	Growth	Growth rate of operating income.
Return on assets	ROA	Net profit/Total assets.
Operating cash flow	OCF	Operating cash/total assets.
Book-to-market ratio	BM	Corporate book value/market value.
CEO duality	Duality	An indicator variable that equals 1 if the chairman of the board of directors and the CEO of an enterprise is the same person, and 0 otherwise.
Ownership concentration	Top5	Share proportion of the top five shareholders.
Equity balance	Top5_1	Share proportion of the second to fifth largest shareholders/Share proportion of the largest shareholder.
Board size	Board	The number of directors.
Proportion of independent directors	Indirector	The ratio of independent directors to all directors.
Firms’ state ownership	SOE	An indicator variable that equals 1 for state-owned firms, and 0 otherwise.
Regional economic development level	GDP	GDP per capita (in ten thousand CNY) in the province where an enterprise is located.

**Table 2 ijerph-19-09812-t002:** Descriptive statistics.

Variable	N	Mean	S.D.	Min	P50	Max
ER	6203	3.3180	0.8880	0.6930	3.3670	5.4160
Dige	6203	0.9460	1.9530	−0.9500	0.2760	8.5140
Size	6203	22.1820	1.2960	19.9270	21.9620	26.1380
Growth	6203	0.1580	0.3450	−0.4930	0.1070	2.5920
ROA	6203	0.0450	0.0570	−0.1880	0.0400	0.2130
OCF	6203	0.0570	0.0660	−0.1300	0.0560	0.2410
BM	6203	0.6270	0.2450	0.1210	0.6280	1.1560
Duality	6203	0.2480	0.4320	0.0000	0.0000	1.0000
Top5	6203	54.6560	15.2460	19.9280	55.1240	89.8400
Top5_1	6203	0.6710	0.5800	0.0230	0.5100	2.6840
Board	6203	8.8420	1.7440	5.0000	9.0000	15.0000
Indirector	6203	37.0340	5.0900	33.3300	33.3300	57.1400
SOE	6203	0.4110	0.4920	0.0000	0.0000	1.0000
GDP	6203	6.3060	2.7070	2.3150	5.8830	14.0210

**Table 3 ijerph-19-09812-t003:** Digital economy development and corporate energy conservation and emission reduction.

	(1)	(2)	(3)	(4)	(5)
	ER	ER	ER	ER	ER
Dige	0.0467 ***	0.0562 ***	0.2075 ***	0.0626 ***	−0.0655
	(3.4811)	(3.2707)	(3.2066)	(4.8132)	(−1.4664)
Size		0.0930 **	−0.1876 *	0.0955 **	0.4388 **
		(2.3384)	(−1.7370)	(2.0995)	(2.2062)
Growth		−0.0233	0.0181	−0.0267	0.0330
		(−0.9652)	(0.2486)	(−0.9449)	(0.7583)
ROA		0.0724	−1.1584	0.0990	1.1899 *
		(0.4019)	(−1.6851)	(0.5123)	(1.7698)
OCF		0.1953 *	−0.2200	0.2133 *	−0.1628
		(1.6760)	(−0.4020)	(1.7612)	(−0.2128)
BM		0.1315 **	−0.1356	0.1357 **	−0.2914
		(2.1933)	(−0.3844)	(2.0643)	(−0.9199)
Duality		−0.0478 *	−0.0489	−0.0272	−0.3849 **
		(−1.6950)	(−0.3475)	(−0.9676)	(−2.5631)
Top5		0.0003	−0.0020	0.0012	−0.0032
		(0.1524)	(−0.3405)	(0.6576)	(−0.6588)
Top5_1		−0.0585 *	−0.0184	−0.0761 **	0.0483
		(−1.7643)	(−0.1402)	(−2.2038)	(0.4072)
Board		0.0124	0.0283	0.0148	−0.0015
		(1.0630)	(1.2715)	(1.0834)	(−0.0927)
Indirector		−0.0014	−0.0231 *	0.0019	−0.0106
		(−0.4455)	(−1.7917)	(0.6122)	(−1.3270)
SOE		0.0911	0.2826	0.1026	−0.3381
		(0.9737)	(1.1382)	(1.0032)	(−1.0606)
GDP		−0.0226	−0.1059 *	−0.0210	0.0409
		(−1.0990)	(−1.8202)	(−0.8141)	(0.8056)
Constant	3.2734 ***	1.1929	8.9714 ***	0.8971	−5.3900
	(251.4223)	(1.2473)	(3.2825)	(0.8644)	(−1.2392)
Firm/Year	YES	YES	YES	YES	YES
N	6203	6203	338	5445	417
R-squared	0.8228	0.8252	0.6987	0.8301	0.8089

Note: ***, **, * represent the significance levels of 1%, 5%, and 10%, respectively; In parentheses are the *t*-values of the two-sided test corresponding to the cluster-robust standard errors of cities, the same as in the table below.

**Table 4 ijerph-19-09812-t004:** Instrumental variable method.

	(1)	(2)
	The First Stage	The Second Stage
	Dige	ER
Dige		0.1585 *
		(1.9169)
Post_Intnet	0.0008 ***	
	(7.8247)	
Aver_Dige	0.5135 ***	
	(2.7451)	
Size	−0.0241	0.1282 ***
	(−0.0810)	(3.1684)
Growth	−0.0000	−0.0277
	(−0.0000)	(−1.0944)
ROA	0.3068	0.0060
	(1.4013)	(0.0336)
OCF	−0.1245	0.2344 *
	(−1.0918)	(1.7951)
BM	0.2608	0.0529
	(1.6118)	(0.8494)
Duality	−0.0119	−0.0495 *
	(−0.4757)	(−1.7459)
Top5	0.0017	0.0013
	(1.2003)	(0.9186)
Top5_1	−0.0098	−0.0638 **
	(−0.4405)	(−2.0113)
Board	0.0013	0.0141
	(0.1743)	(1.3007)
Indirector	−0.0016	−0.0017
	(−0.8190)	(−0.5419)
SOE	−0.0300	0.0653
	(−0.5026)	(0.6669)
GDP	0.2510 ***	−0.0614 *
	(2.6288)	(−1.8871)
Firm/Year	YES	YES
N	5651	5651
Kleibergen–Paap rk LM statistic *p*-value	0.0123	
Cragg-Donald Wald F statistic	470.53	
Stock-Yogo weak ID test: 10% maximal IV size	19.93	
Hansen J statistic *p*-value	0.3710	

Note: ***, **, * represent the significance levels of 1%, 5%, and 10%, respectively; In parentheses are the *t*-values of the two-sided test corresponding to the cluster-robust standard errors of cities, the same as in the table below.

**Table 5 ijerph-19-09812-t005:** DID regression results.

	(1)	(2)	(3)
	DID	PSM-DID	Balance
	ER	ER	ER
Treat × Post	0.1541 ***	0.1540 ***	
	(3.4473)	(3.3445)	
Pre3			−0.0100
			(−0.3029)
Pre2			−0.0138
			(−0.2371)
Pre1			0.0969
			(1.6496)
Current			0.1357 **
			(2.1011)
After1			0.1741 ***
			(2.9175)
After2			0.1965 ***
			(3.2609)
After3			0.1900 ***
			(2.9112)
Size	0.0895 **	0.0892 **	0.0888 **
	(2.2414)	(2.1959)	(2.2216)
Growth	−0.0244	−0.0254	−0.0246
	(−1.0438)	(−1.0780)	(−1.0456)
ROA	0.1005	0.0728	0.1067
	(0.5679)	(0.4073)	(0.6038)
OCF	0.2004 *	0.2206 *	0.2011 *
	(1.7418)	(1.9072)	(1.7501)
BM	0.1333 **	0.1361 **	0.1336 **
	(2.1440)	(2.1342)	(2.1363)
Duality	−0.0480 *	−0.0529 *	−0.0481 *
	(−1.7548)	(−1.9592)	(−1.7692)
Top5	0.0004	0.0005	0.0003
	(0.2143)	(0.3214)	(0.1964)
Top5_1	−0.0538	−0.0522	−0.0512
	(−1.6505)	(−1.6062)	(−1.5844)
Board	0.0131	0.0134	0.0129
	(1.1152)	(1.0675)	(1.1008)
Indirector	−0.0014	−0.0018	−0.0014
	(−0.4497)	(−0.5036)	(−0.4432)
SOE	0.0950	0.1074	0.0988
	(1.0009)	(1.1254)	(1.0374)
GDP	−0.0131	−0.0137	−0.0151
	(−0.8377)	(−0.8186)	(−0.9990)
Constant	1.2169	1.2228	1.2373
	(1.3064)	(1.2759)	(1.3292)
Firm/Year	YES	YES	YES
N	6203	6145	6203
R-squared	0.8261	0.8261	0.8264

Note: ***, **, * represent the significance levels of 1%, 5%, and 10%, respectively; In parentheses are the *t*-values of the two-sided test corresponding to the cluster-robust standard errors of cities, the same as in the table below.

**Table 6 ijerph-19-09812-t006:** Robustness tests.

	(1)	(2)	(3)	(4)	(5)	(6)	(7)
	ER	ER	ER	ER	ER	SO_2_	ER_Ind
Dige	0.0457 **	0.0828 ***	0.0679 ***	0.0562 **	0.0562 *	−0.0494 ***	0.0502 ***
	(2.0039)	(3.5613)	(4.3652)	(2.3786)	(2.0336)	(−3.1586)	(2.9767)
Dige × Policy	0.0117						
	(0.9181)						
Size	0.0920 **	0.1072 **	0.0905 **	0.0930 **	0.0930 **	0.1048	0.0814 **
	(2.3177)	(1.9940)	(2.2071)	(2.2400)	(2.6716)	(0.9098)	(1.9815)
Growth	−0.0227	−0.0107	−0.0195	−0.0233	−0.0233	−0.0109	−0.0145
	(−0.9444)	(−0.3817)	(−0.8022)	(−1.1181)	(−0.7522)	(−0.2549)	(−0.5877)
ROA	0.0818	0.3497	0.0414	0.0724	0.0724	−0.0070	0.0829
	(0.4472)	(1.2140)	(0.2235)	(0.3499)	(0.5382)	(−0.0314)	(0.4520)
OCF	0.1962 *	0.1204	0.1979 *	0.1953	0.1953 *	−0.0375	0.1936
	(1.6870)	(0.7563)	(1.7030)	(1.5037)	(1.9454)	(−0.4470)	(1.5507)
BM	0.1282 **	0.1534	0.1181 *	0.1315 *	0.1315	−0.3042	0.0937
	(2.1243)	(1.3266)	(1.9025)	(1.7956)	(1.8388)	(−0.9178)	(1.5305)
Duality	−0.0476 *	−0.0290	−0.0418	−0.0478 *	−0.0478	0.0117	−0.0472 *
	(−1.6935)	(−0.7224)	(−1.4514)	(−1.6759)	(−1.8221)	(0.4468)	(−1.6637)
Top5	0.0002	0.0002	0.0012	0.0003	0.0003	−0.0005	0.0005
	(0.1057)	(0.1099)	(0.7329)	(0.1517)	(0.1726)	(−0.4072)	(0.3085)
Top5_1	−0.0580 *	−0.0237	−0.0662 **	−0.0585	−0.0585	0.0480	−0.0482
	(−1.7525)	(−0.4288)	(−2.0100)	(−1.6213)	(−1.7972)	(0.9020)	(−1.4747)
Board	0.0125	−0.0115	0.0117	0.0124	0.0124	0.0151	0.0079
	(1.0689)	(−0.7243)	(0.9851)	(1.1609)	(1.0307)	(0.8675)	(0.6568)
Indirector	−0.0015	−0.0056	−0.0013	−0.0014	−0.0014	−0.0039	−0.0012
	(−0.4791)	(−1.5528)	(−0.4212)	(−0.4144)	(−0.4309)	(−0.5739)	(−0.4321)
SOE	0.0903	0.0109	0.0774	0.0911	0.0911	−0.0706	0.0330
	(0.9690)	(0.0890)	(0.8494)	(1.0476)	(1.1532)	(−0.9451)	(0.3365)
GDP	−0.0305	−0.0579	−0.0175	−0.0226	−0.0226	0.0178	−0.0108
	(−1.5040)	(−1.1557)	(−0.7819)	(−1.2577)	(−0.9945)	(0.8140)	(−0.5185)
Constant	1.2765	1.2432	1.1702	1.1929	1.1929	−2.1719	−1.8803 *
	(1.3585)	(0.9078)	(1.2260)	(1.2613)	(1.3955)	(−0.9557)	(−1.9398)
Firm/Year	YES	YES	YES	YES	YES	YES	YES
Industry/City	NO	NO	YES	NO	NO	NO	NO
N	6203	3479	6202	6203	6203	323	6203
R-squared	0.8253	0.8378	0.8290	0.8252	0.8252	0.8467	0.7489

Note: ***, **, * represent the significance levels of 1%, 5%, and 10%, respectively.

**Table 7 ijerph-19-09812-t007:** Control ESG investment and QE policy.

	(1)
	ER
Dige	0.0531 **
	(2.5055)
Size	0.0914 **
	(2.2732)
Growth	−0.0215
	(−0.8795)
ROA	0.0013
	(0.0065)
OCF	0.1945 *
	(1.6748)
BM	0.1308 **
	(2.1726)
Duality	−0.0499 *
	(−1.7678)
Top5	0.0002
	(0.1338)
Top5_1	−0.0571 *
	(−1.7149)
Board	0.0111
	(0.9857)
Indirector	−0.0017
	(−0.5571)
SOE	0.1165
	(1.2133)
GDP	−0.0229
	(−1.0660)
ESG	0.0121
	(0.9480)
QE	−0.5822 ***
	(−8.8788)
Constant	0.2430
	(0.2824)
Firm/Year	YES
N	6144
R-squard	0.825

Note: ***, **, * represent the significance levels of 1%, 5%, and 10%, respectively.

**Table 8 ijerph-19-09812-t008:** Coefficient stability analysis.

$\delta =-1,R_{max}=0.83$	$\delta =-1,R_{max}=0.9$	$\delta =-1,R_{max}=0.95$	$\delta =-1,R_{max}=1$
β=0.05098	β=0.02927	β=0.02583	β=0.02407

**Table 9 ijerph-19-09812-t009:** Analysis of impact mechanism.

	(1)	(2)	(3)
	Innovation	SA	Compete
Dige	0.0250 **	−0.0083 **	−0.0039 **
	(2.5764)	(−2.5324)	(−2.0464)
Size	0.0010	0.0340 ***	0.0033
	(0.0702)	(5.3304)	(1.0624)
Growth	−0.0088	0.0049 *	−0.0008
	(−1.0697)	(1.8800)	(−0.5603)
ROA	0.0785	−0.0079	0.0253
	(0.9654)	(−0.4261)	(1.0789)
OCF	0.0045	0.0150	0.0097
	(0.0587)	(1.3313)	(1.0627)
BM	0.0292	−0.0154 *	0.0037
	(0.8809)	(−1.9575)	(1.0830)
Duality	−0.0020	−0.0022	−0.0015
	(−0.1030)	(−0.7919)	(−0.7917)
Top1_5	−0.0004	−0.0009 ***	0.0001
	(−0.6677)	(−3.7208)	(1.4832)
Top5_1	−0.0114	0.0019	0.0017
	(−0.4954)	(0.5879)	(0.6621)
Board	0.0010	−0.0008	0.0006
	(0.1338)	(−0.6687)	(0.8550)
Indirector	0.0025	−0.0001	−0.0002
	(1.3660)	(−0.2987)	(−0.8682)
SOE	−0.0540	−0.0026	0.0056
	(−0.7873)	(−0.2680)	(0.9940)
GDP	0.0004	0.0027	−0.0003
	(0.0411)	(1.1457)	(−0.2592)
Constant	−0.0313	3.0557 ***	0.0010
	(−0.1128)	(20.3878)	(0.0142)
Firm/Year	YES	YES	YES
N	6203	6203	6201
R-squared	0.7565	0.9835	0.9270

Note: ***, **, * represent the significance levels of 1%, 5%, and 10%, respectively; In parentheses are the *t*-values of the two-sided test corresponding to the cluster-robust standard errors of cities, the same as in the table below.

**Table 10 ijerph-19-09812-t010:** Heterogeneity between regional economic development level and local government financial pressure.

	(1)	(2)	(3)	(4)
	Developed Region	Less developed Region	High Financial Pressure	Low Financial Pressure
	ER	ER	ER	ER
Dige	0.0513 **	0.0571	0.0624	0.0614 ***
	(2.6010)	(1.5156)	(0.6992)	(2.8931)
Size	0.0589	0.1253 **	0.1230 **	0.0407
	(1.1781)	(2.0931)	(1.9738)	(0.7790)
Growth	−0.0053	−0.0388	−0.0300	−0.0096
	(−0.2464)	(−1.0363)	(−0.8386)	(−0.2888)
ROA	0.3227	−0.2041	−0.1207	0.2513
	(1.2090)	(−0.7936)	(−0.4560)	(0.8951)
OCF	0.0101	0.3742 **	0.2895	0.1078
	(0.0727)	(2.0636)	(1.5289)	(0.7462)
BM	0.1033	0.1643 *	0.1422	0.1197
	(1.4952)	(1.7212)	(1.5165)	(1.5471)
Duality	−0.0405	−0.0268	−0.0230	−0.0586
	(−1.0668)	(−0.6658)	(−0.5175)	(−1.5113)
Top5	0.0002	−0.0002	−0.0001	0.0015
	(0.1264)	(−0.0613)	(−0.0418)	(0.7777)
Top5_1	0.0053	−0.1062 **	−0.0737	−0.0360
	(0.1196)	(−2.1154)	(−1.4107)	(−0.8746)
Board	−0.0013	0.0236 *	0.0213	0.0021
	(−0.0635)	(1.7384)	(1.5352)	(0.0961)
Indirector	−0.0040	0.0012	0.0032	−0.0054
	(−0.8042)	(0.3282)	(0.9009)	(−0.9396)
SOE	0.0921	0.0818	0.1005	0.0685
	(0.7086)	(0.6474)	(0.7858)	(0.5709)
GDP	0.0032	−0.0305	0.0025	−0.0255
	(0.1253)	(−0.7206)	(0.0565)	(−0.7464)
Constant	1.9258	0.3774	0.2095	2.4953 *
	(1.6180)	(0.2663)	(0.1437)	(1.7940)
Firm/Year	YES	YES	YES	YES
N	3098	3057	3000	3060
R-squared	0.8345	0.8237	0.8237	0.8342

Note: ***, **, * represent the significance levels of 1%, 5%, and 10%, respectively; In parentheses are the *t*-values of the two-sided test corresponding to the cluster-robust standard errors of cities, the same as in the table below.

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
