# Peer review of "Can Digital Economy Promote Energy Conservation and Emission Reduction in Heavily Polluting Enterprises? Empirical Evidence from China"

_ijerph, 2022, doi:10.3390/ijerph19169812_

Round 1
Reviewer 1 Report
From the overall presentation I would say that interesting research work has been done. The topic is also important for the readers of the journal. However, I have a few more significant challenges with the paper.
The authors should include more clearly in the literature section the research hypotheses.
In “3.2. Model specification and variable definition” the authors should add more information about the “comprehensive index of the digital economy development level”.
Please include the measurement units in the case of all analyzed variables (see table 1).
The discussion of the results remains quite superficial. The authors need to improve the practical and academic implications.
The original contribution of the research has to be presented by focusing on the research results based on the research questions. The empirical data would give more possibilities to improve the theory further.
However, the paper must underline the limits of the research and future work.
The quality of the tables is not sufficient.
The author has to pay attention to references inside the paper as well as the reference list.
[For example, “This paper follows the method of Zhao et al. [49]”...see References “50. Zhao, T.; Zhang, Z.; Liang, S.J.M.W. Digital Economy, Entrepreneurship, and High-Quality Economic Development: Empirical 646 Evidence from Urban China. Management World 2020, 36, 65-76. (In Chinese)”]
Author Response
Reviewer 1’s Comments and Authors’ Responses
Thank you for your careful review of our paper, and for your insightful comments. We have worked to incorporate your suggestions into this version of the manuscript, and we feel that this has greatly enhanced the paper.
Our responses to your specific suggestions are listed below. We provide either a replication or a general heading concerning each of your comments, followed by a discussion of the changes made to the paper to address your concerns.
Reviewer 1 Comments:
From the overall presentation I would say that interesting research work has been done. The topic is also important for the readers of the journal. However, I have a few more significant challenges with the paper.
1) The authors should include more clearly in the literature section the research hypotheses.
Authors’ Response: Thank you very much for this suggestion. We have revised related section and outlined our research hypotheses more clearly (page 5, section 2).
2) In “3.2. Model specification and variable definition” the authors should add more information about the “comprehensive index of the digital economy development level”.
Authors’ Response: We appreciate this suggestion. We have added detailed information about the “comprehensive index of the digital economy development level” in the related section (page 6, section 3.2).
3) Please include the measurement units in the case of all analyzed variables (see table 1).
Authors’ Response: Thank you for this suggestion. Most variables in Table 1 are either unitless ratios, dummy variables, or natural log values. We have added measurement units for corporate total assets and GDP per capital in the definitions for related variables in Table 1.
4) The discussion of the results remains quite superficial. The authors need to improve the practical and academic implications.
Authors’ Response: Thank you for pointing this out. We have added more discussions on practical and academic implications (page 3 in section 1 and pages 19 and 20 in section 6).
5) The original contribution of the research has to be presented by focusing on the research results based on the research questions. The empirical data would give more possibilities to improve the theory further. However, the paper must underline the limits of the research and future work.
Authors’ Response: We appreciate your suggestions and comments. We have added more discussions on contributions, limitations, and future research (page 3 in section 1 and pages 19 and 20 in section 6).
6) The quality of the tables is not sufficient.
Authors’ Response: We appreciate your suggestions and comments. We have revised the format of some tables and will further follow the journal editor’s instructions to adjust table format in accordance with journal requirements.
7) The author has to pay attention to references inside the paper as well as the reference list. [For example, “This paper follows the method of Zhao et al. [49]”...see References “50. Zhao, T.; Zhang, Z.; Liang, S.J.M.W. Digital Economy, Entrepreneurship, and High-Quality Economic Development: Empirical 646 Evidence from Urban China. Management World 2020, 36, 65-76. (In Chinese)”]
Authors’ Response: This was an oversight on our part. We have corrected references inside the paper and the reference list.
We really appreciate your time and efforts to review our manuscript. And we respectfully believe our manuscript is completed.
Reviewer 2 Report
Please refer to the attached referee report.

Author Response
Reviewer 2’s Comments and Authors’ Responses
Thank you for your careful review of our paper, and for your insightful comments. We have worked to incorporate your suggestions into this version of the manuscript, and we feel that this has greatly enhanced the paper.
Our responses to your specific suggestions are listed below. We provide either a replication or a general heading concerning each of your comments, followed by a discussion of the changes made to the paper to address your concerns.
Reviewer 2 Comments:
This paper examines whether digital economy features affect corporate energy conservation and emission reduction for a set of listed enterprises in China. The authors conclude that the answer to the afore- mentioned research question is positive and provide a series of additional tests shedding light on heterogeneous effects.
My overall assessment is that the research question is worthy of consideration for publication given its conceptual and practical relevance for academics and policymakers. Generally speaking, the paper is well executed and the authors do not over-claim the extent of their findings.
Despite the positive outlook, I believe the present version of the paper is silent regarding important macro forces pertaining to the sample period of the study and the authors did not attempt to estimate the magnitude of the corresponding econometric biases that can arise because of such forces. As such, for the paper to be publishable in an academic outlet such as the International Journal of Environmental Research and Public Health, the authors need to (at least) accomplish the following.
|
- Discuss the two relevant market forces: (a)increasing popularity of ESG investing; (b) Massive injection of market liquidity due to quantitative easing policies put forth by major central banks, cite the relevant literature and explain how these variables might affect their main estimates.
Authors’ Response: Thank you very much for these valuable suggestions. We have incorporated discussions on these two relevant market forces in the manuscript and cited relevant literature. Empirically, we have performed an additional test to control the impact of Fed QE policy and corporate ESG investment.
Following Hattori et al. (2016) and Cortes et al. (2022), we control the impact of QE policies using event-study approach. Specifically, we set dummy variables based on the timing of the Fed's QE policy (Luck and Zimmermann, 2020). We do not use the log-size of the Fed's balance sheet as a proxy for QE policy since our panel data should control for extremely important time fixed effects.
In addition, we use enterprise ESG ratings as a proxy for ESG investment due to the unavailability of corporate ESG investment data. The rationality of this approach is that only companies that invest more in ESG can obtain better rating. ESG rating scores are measured by the ESG index developed by Sino-Securities Index Information Service (Shanghai) Co. Ltd, which is widely recognized by the industry and academia (Lin et al., 2021).
As shown in Table 7, our core explanatory variables are still significantly positive after incorporating QE policy and ESG rating into the benchmark model for testing.
2)
|
Implement a simple method (based on Oster, 2019) to estimate the magnitude of potential estimation biases.
Authors’ Response: Thank you very much for this suggestion. We follow the method of Oster (2019) to estimate the magnitude of potential estimation biases. Under the condition of , we estimate the value range of β by changing the value range of ). If the value range of β does not include 0, it means that the coefficient is stable. As shown in Table 8, the value range of Beta is 0.02407 to 0.05098, excluding 0, indicating that our results are robust, and the core conclusions of this paper are not too biased due to the problem of missing variables.
3) Additional comment. The authors should be more specific in their sample description, particularly concerning the definition of “heavily polluting industries.” Readers may have an idea of what industries the authors are referring to, but an academic paper needs to be self-contained regarding the description of the empirical analysis.
Authors’ Response: Thank you for pointing this out. We have added more specific descriptions about the sample and the definition for “heavily polluting industries” (page 5, section 3.1).
We really appreciate your time and efforts to review our manuscript. And we respectfully believe our manuscript is completed.